# Effect of Particle Size and Morphology of Siliceous Supplementary Cementitious Material on the Hydration and Autogenous Shrinkage of Blended Cement

**DOI:** 10.3390/ma16041638

**Published:** 2023-02-16

**Authors:** Xiaochuan Xu, Yunqi Zhao, Xiaowei Gu, Zhenguo Zhu, Fengdan Wang, Zaolin Zhang

**Affiliations:** 1School of Resources and Civil Engineering, Northeastern University, Shenyang 110819, China; 2Science and Technology Innovation Center of Smart Water and Resource Environment, Northeastern University, Shenyang 110819, China; 3School of Mathematics and Physical Sciences, University College London, London WC1E 6BT, UK

**Keywords:** iron ore tailing powder, mechanical activation, particle morphology, hydration reaction, autogenous shrinkage

## Abstract

Supplementary cementitious material (SCM) plays an important role in blended cement, and the effect of the particle size and morphology of siliceous supplementary cementitious material on hydration should not be ignored. In this study, 0.5 h and 1 h of wet grinding was applied to pretreat iron ore tailing powder (TP), and the divergence in pozzolanic behavior and morphology were investigated. Then, the treated TPs were used to replace the 30% cement contents in preparing blended cementitious paste, and the impact mechanism of morphology on performance was studied emphatically. M, the autogenous shrinkages of pastes were tested. Finally, hydration reaction kinetics was carried out to explore the hydration behavior, while X-ray diffraction (XRD) and thermogravimetric analysis (TGA) were used to characterize the hydration product properties, respectively. Meanwhile, microscopy intrusion porosimetry (MIP) was also carried out to characterize the pore structures of hardened specimens. Results indicated that wet grinding has a dramatic effect on particle size and morphology, but hardly affects the phase assemblages and pozzolanic reactivity of TP, while the particle shape of TP changes from sub-circular to clavate and, finally, back to sub-circular. The results of hydration reaction kinetics, representing the morphology of particles, had a significant effect on hydration rate and total heat, and compared with the sub-circle one, the clavated particle could inhibit the hydration procedure. With the increasing grinding time, the compressive strength of cementitious paste was increased from 17.37% to 55.73%, and the micro-pore structure became denser; however, the autogenous shrinkage increased.

## 1. Introduction

The concrete industry is an important industry all over the world, and ordinary Portland cement (OPC) is a cementitious material that is widely acknowledged due to its excellent properties regarding operability and mechanics [1]. Although OPC shows many extraordinary advantages, its disadvantages are also clear. The production of OPC consumes a lot of natural resources such as limestone and clay, which are both non-renewable. Additionally, the high energy consumption required for the decomposition of limestone and the direct emission of carbon dioxide during this process also cause serious environmental problems, such as contributing to the greenhouse effect [2]. Statistically speaking, the cement industry contributes about 7% of total carbon dioxide emissions [3,4].

Due to the serious environmental problems posed by the production of cement, a series of green supplementary cementitious materials (SCMs) are being studied and developed to partially replace cement and thus decrease environmental pollution [5]. SCMs can be mainly classified into two categories: natural pozzolanic minerals (such as pozzolana, kaolin or metakaolin, and calcined clay) [6,7,8] and industrial by-products [9,10,11]. Considering their relatively low cost and the benefits to environmental protection, industrial by-products with high pozzolanic activity and/or hydration capacity, such as steel slag [12], slag [13,14], fly ash [15,16], and silica fume [17], are very suitable to be used in the development of cementitious composites as a substitute for cement. SCMs mainly contribute by filling pores, consuming calcium hydroxide, producing hydrated calcium silicate, and reducing the interface transition zone, which improves not only the mechanical behaviors but also the workability and durability of cementitious composites [18,19,20,21]. For example, slag, a by-product of the metal-smelting process that is composed of a large number of amorphous mineral contents, exhibits very high pozzolanic reactivity and improves the mechanical strength and compactness of cementitious composites [22,23]. Its high reactivity makes slag the most widely used SCM both in cementitious and alkali-activated materials [13,24]. However, with the rapid development of the concrete industry, these high-pozzolanic SCMs are in short supply, especially in China, meaning the prices of SCMs tend to rise.

Iron tailing, as a main industrial solid waste in China, occupies a lot of land and causes many environmental problems [25,26]. The chemical composition of iron tailing varies with different regions, but silicon is one of the main elements [25]. The silicon content of iron tailing can account for more than 40~60%, especially in Northeastern China, where the silicon content can reach 70% [26]. The rich content of quartz makes the coarse tailing suitable for use as an aggregate in cemented paste backfill (CPB) and cementitious composites [27,28]. Contrastingly, the high content of quartz also makes the fine tailing (particle size < 100 μm) show very low pozzolanic activity. Hence, there are many scholars who have studied the utilization of iron tailing powder (TP) as a replacement for cement to develop cementitious composites. Han et al. [29,30,31] carried out many studies on silicon-rich iron tailing powder, which indicated that iron tailing shows very low pozzolanic activity and mainly fills micropores. Zhao et al. [32] discussed the hydration behavior of cement paste blended with iron tailing and found that both finer particle size and higher temperature had a positive effect on the pozzolanic behavior of iron tailing cement paste. Considering the low pozzolanic activity, there are also some scholars who have studied the pretreating method to improve the pozzolanic activity of TP. Grinding, as a cheap and efficient mechanical method, is often used to increase the specific surface area and activate SCMs such as fly ash [33] and steel slag [34]. Beyond the effect of particle size, previous studies have also reported that the mechanical process has a significant effect on the phase of silicon, during which the quartz can be translated into amorphous silica [35,36]. Although the effect of mechanical activation on the pozzolanic behavior of iron tailing has already been studied, modifications in morphology during the grinding process are rarely discussed. Meanwhile, study of the early properties of cement tailing powder-based mortar, especially regarding hydration behavior and autogenous shrinkage, is also limited. Hydration behavior and autogenous shrinkage at an early age dramatically affect the micro-pore structure and strength of cementitious materials at both early and later ages. A change in the morphology of particles also has a significant effect on the hydration of cement and the packing density of materials, and finally influences the mechanical behaviors and durability of cementitious composites [37]. Therefore, it is necessary to study and discuss the influence of morphology of SCM particles on the early properties of cementitious composites.

Although inexpensive and simplified, the experimental phenomenon particle aggregation gradually becomes remarkable as a result of increasing surface energy when grains are ground to a certain fine size. Considering the particle aggregation phenomenon caused by electrostatic charge during the dry grinding, in this study a wet grinding method was used to gain a finer particle size distribution and mechanically activate the iron tailing effectively. The properties, especially the morphology, of TP after grinding was analyzed, and the effect of mechanically activated TP on the early behavior of cemented paste was studied. Firstly, a series of tests, including particle size distribution (PSD), X-ray diffraction (XRD), and scanning electron microscope (SEM), were carried out to characterize the iron tailing powder before and after grinding for 0.5 h or 1 h. Then, to prepare the cemented paste, three tailing powders (raw, 0.5 h of grinding, and 1 h of grinding) were used to replace the 30% cement contents, while the water/binder ratio as fixed at 0.5. Finally, the specimens were cured for 3d, and a mechanical test, hydration product analysis, micropore structure analysis, and autogenous shrinkage were carried out to assess the effect of mechanically activated TPs on the early behavior of cemented paste.

## 2. Materials and Methods

### 2.1. Materials

A P.O. 42.5 cement (OPC), provided by Xiaoyetian Co., Ltd. (Dalian, China), was used as the main cementitious binder. Iron ore tailing powder (TP), collected from Waitou Mountain Iron Mine (Benxi, China), was mechanically activated and studied as an SCM used to replace 30% (by weight) of the OPC, following Chinese Standard GB/T 12957-2005. The chemical compositions and specific surface areas of the raw materials are shown in Table 1, while the particle size distributions of the raw materials are represented in Figure 1.

### 2.2. Procedure Used for the Mechanical Activation of TP

It has been concluded that a mechanical force can affect solid particles physically and chemically, where the particle sizes, specific surface areas, and mineralogical properties may be influenced and transformed as well [38]. Hence, a mechanical method was carried out to improve the pozzolanic behavior of iron ore tailing, which is much more ecological and environmental than other treatment methods such as calcination. Dry grinding is often seen as an effective method for activating the pozzolanic behavior of different SCM, including iron tailing. However, although inexpensive and simplified, the experimental phenomenon named particle aggregation gradually becomes remarkable as a result of increasing surface energy when grains are ground to a certain fine size.

On the contrary, many researchers and studies have reported that, when raw materials and water are ground together, a process named wet grinding, a finer particle size and very low particle aggregation are attained [34,39]. Moreover, the wet environment is also conducive to the dissolution of reactive mineral. Hence, in this study, TP was ground in wet conditions using a planetary ball mill. A group of zirconia balls matching the ratio 20 mm:16 mm:12 mm:4 mm = 1:4:6:2, by weight, was used as grinding media. The following fixed experimental parameters were employed: a material/ball ratio (mass ratio of TP to grinding media by weight) of 0.5, a water/material ratio (mass ratio of water to TP by weight) of 0.4, and two different grinding periods of 0.5 h and 1 h.

### 2.3. Mix Design

To assess the effect of the mechanical activation of TPs, series groups of cemented paste were made. For the controlled specimen (named PC), OPC was used as the unique cementitious material, and the water-to-cement binder ratio (w/b) was 0.5. For other experimental groups, different mechanically activated TPs are used to replace the 30 wt% cement content, and w/b was fixed at 0.5, following the Chinese Standard GB/T 17671-2021.

### 2.4. Mixing Procedure and Sample Preparation

A NJ-160 mixer possessing two rotating speeds of slow (140 ± 5 rpm) and high (285 ± 5 rpm) was used to prepare the cementitious paste. Firstly, all pre-dried cementitious binders (OPC or OPC and TP) were poured into the container and mixed for 3 min, ensuring the uniformity of particles; then the water was added into the mixture and mixed for 2 min. During both of the above two stages, the mixtures were rotated at a low speed. Finally, the mixture was mixed at a high speed for another 3 min.

### 2.5. Testing Methods

#### 2.5.1. Air Content

To assess the influence of mechanically activated TP on the air bubble and pore structure of cementitious composites, an air content test was carried out using a measurement instrument suggested by Chen et al. [40]. The air contents of fresh pastes were measured following ASTM C231-17a using a Type-B meter and calculated using Equation (1).
*A_s_* = *A*_1_ − *G*(1)
where *A_s_* is the air content of each fresh mixture (%), *A*_1_ is the air content shown by the Type-B meter (%), and *G* is the correction factor (%). A triplicate testing procedure was performed and the average determined the air content value.

#### 2.5.2. Setting Time

The setting time of the fresh slurry was tested using the ZKS-100 Vicat apparatus according to the methods reported by Zhao et al. [13] and Wu et al. [41]. The time at which needle penetration depth reached 34 mm was defined as the initial setting time, the time at which it reached 3 mm was defined as the final setting time.

#### 2.5.3. Heat of Hydration

To study the effect of mechanically activated TP on the reaction kinetics of early hydration, a Thermometrics TAM Air isothermal conduction calorimeter was used to monitor the heat of hydration for 3 days, at an ambient temperature of 20 °C.

#### 2.5.4. Compressive Strength

After mixing, different curing treatments for specimens were employed, dominated by series experimental tests. For compressive strength test, the fresh mixture was poured into a mold with a cube size of 40 mm × 40 mm × 40 mm, left to stand for 24 h at a temperature of 20 ± 2 °C and a relative humidity of 65 ± 2%, and then de-molded. Then, the hardened mixture was cured in the curing room at a temperature of 20 ± 2 °C and a relative humidity of 95 ± 2%. After curing for 1, 3, 7, or 28 d, the specimens are carried out and tested, and the mean values resulting from triplicate tests were determined to be the compressive strengths of the mixtures.

#### 2.5.5. Hydration Product Analysis

A slice was cut from the hardened paste after being cured for 28 d, which was impregnated with isopropanol to stop the hydration of unreacted cement. Then, the slice was dried under vacuum at a temperature of 40 °C for 24 h and ground into a powder. After this, the mineralogical and quantitative properties of the hydration products were determined through XRD and DTG analyses at room temperature. The XRD patterns of hardened pastes were collected from 5° to 50° 2θ at a scanning speed of 5°/min, and DTG curves were gained at temperatures ranging from 30 °C to 1000 °C with a heating rate of 15 °C/min.

#### 2.5.6. Mercury Intrusion Porosimetry (MIP)

To evaluate the micropore structure of samples, a Micromeritics Mercury Porosimeter (AutoPore IV-9500) with a test range of 3 nm to 360 μm was used. A slice cut from a specimen which had been cured for 28 d was tested to determine its micropore structure. The slice was impregnated with isopropanol and dried for 24 h.

#### 2.5.7. Autogenous Shrinkage

A prismatic mold of 25 mm × 25 mm × 280 mm with a gauge length of 250 mm was used to perform the autogenous shrinkage test of the cementitious mixture based on the Standard ASTM C490/C490M. The testing method was mentioned by Wang et al. [42]. Once it had reached the final setting time, the specimen was de-molded carefully, and the initial comparator reading was taken immediately. Afterwards, the specimen was cured in an environment with a temperature of 20 ± 2 °C and a relative humidity of 65 ± 5%. During the testing, the comparator recorded a measurement every 15 min for 3 d. Afterwards, the indication was recorded every 12 h for 28 d.

#### 2.5.8. Scanning Electron Microscope (SEM) Test

In this study, to quantitatively describe and compare the morphology of TPs during the grinding process, a Scanning Electron Microscope (SEM) machine was used to take nano-micron pictures, and an Image Pro Plus software (IPP) was used to draw and quantify the morphology. The determination of the parameters and calculating process of the calculation is shown in Figure 2, as is a quantitative date representing the graphic and dimensional properties of particles. The specific procedure is detailed as follows.

Firstly, a group of TP powders were taken out and dispersed in an ethanol atmosphere, ensuring that all grains were set on the pallet individually and the dimension of the outline was clear. Then, a micro or nano photo of a particle was taken and translated into a two-dimensional graph, where the geometric size and morphological shape could be characterized clearly. An IPP software was carried out to collect the outline of the particle and a 2D digitized pattern was recorded. Finally, based on exported dimensional data of the outline, two different parameters were collected and calculated to quantitatively describe the morphological properties of TP particles:

(a) Roundness: a circle with the same perimeter was created. The roundness of the particle was defined as the area ratio of circle to particle, for which the value must be higher than 1. The closer the shape is to a circle, the closer the roundness value is to 1; contrarily, the further away the shape from a circle, the higher the roundness value, indicating that the shape of this particle is much more angular or irregular.

(b) L/W ratio: a circumscribed rectangle with the minimum area was created, and the Length (L) and Width (W) were defined as the sizes of the long and short sides of the rectangle, respectively. The ratio between L and W (L/W) was also calculated at the same time. A value of L/W that is far from 1 means that the particles tend to be slender.

## 3. Result and Discussion

### 3.1. Properties of Wet-Ground TPs

#### 3.1.1. Particle Size and Specific Surface Area

The particle size distributions and physical properties of TPs before and after grinding are shown in Figure 3 and Table 2. Unground TP, TP ground for 0.5 h, and TP ground for 1h are referred to as RAWTP, 0.5HTP, and 1.0HTP, respectively.

During the first 30 min (from RAWTP to 0.5HTP), the particle size of TP decreases dramatically, and the dominant geometric size of grains decreased from about 60 μm to 20 μm. The specific surface areas of TPs also substantially increase from 0.4437 to 1.6596 m^2^·g^−1^, by almost four times. This experimental phenomenon has also been reported by previous studies focusing on the mechanism of mechanical activation, where it has been found that the grinding method affects the particle size distribution and specific surface area of materials intuitively and predominantly [43,44]. However, during the second 30 min (from 0.5HTP to 1.0HTP), the decline in particle size is much smaller. The dominant particle size decreases from 20 μm to 16 μm, while the specific surface area increases by merely 51.46% (from 1.6595 to 2.5136 m^2^·g^−1^). This result indicates that the efficiency of mechanical activation substantially declines with a further activation duration, especially for the fourth stage of grinding (1.5HTP to 2.0HTP), where the particle size and specific surface area scarcely change. This phenomenon can be explained as the effect of continuous change in particle size on the grinding behavior. With further grinding time, the difference in diameter between grinding ball and TP grains becomes more serious. The mechanism of ball grinding is considered to be the extrusion and stripping force of the grinding media. A finer particle size is equivalent to a lower probability of contact between grinding media and material. Meanwhile, as a potential reason, the minimum dimension for mineralogical structure remaining stable is definitive and hard broken with an increasing grinding time. Summarily, a permanently incremental grinding behavior would not improve the activation result, but dramatically increase the cost. As an experimental result, the most efficient and reasonably economical activation time is 1 h, and the properties of RAWTP, 0.5HTP, and 1.0HTP will be discussed consequently.

#### 3.1.2. Grinding Mechanism

From Figure 3 and Table 2, there are still some interesting phenomena which should be noted. During the first 30 min (RAWTP to 0.5HTP), the dimensional sizes of TP grains change gradually; both the decrease in the number of coarser particles and increase in the number of finer particles appear to be significant. This indicates that the TP particles are mainly crushed in half or into pieces under the force of extrusion and stripping (Figure 4a). However, during the second 30 min (0.5HTP to 1.0HTP), the general particle size distribution does not change dramatically, but the proportion between 0.7 and 1 μm increases by almost 80%, while other sections change inconspicuously. This indicates that the functional effect of mechanical activation on ground grains (i.e., TPs) is not an extrusion or stripping force, but the friction between media and powder, or between powders. Therefore, the dominant variation during this process is the sensitive changes in grain edge (Figure 4b).

#### 3.1.3. Morphology

For all powdery or granular materials, including cement, SCMs, and aggregate et al., the morphological properties have an apparent and complex impact on the hydrational, mechanical, and durable behaviors of cemented composites. Hence, an accurate and attainable method to quantitatively characterize the particle shape is essential. A 3D size evaluation of particles could characterize the grain effectively. CT or X-CT is a commonly used technology for spatial, dimensional quantification of the morphological properties of rock grains, agricultural grains, and sand, for example [45,46,47]. The most prominent advantages of CT technology are the fact that it is 3D and its meticulousness, as well as its digital characterization. However, the expensive cost and overly complicated operation process make it hard to apply to the constructed site; thus it is mainly used during experimental study in a laboratory. On the contrary, a 2D characterization, i.e., complanate characterization, is much more convenient and economical compared to the CT method. Roundness and angularity are frequently assessed parameters. For example, Wang et al. [48] used Form2D values and angularity to quantify the form of recycled construction and demolition waste, applicating in ultra-high performance concrete.

Figure 5 and Table 3 show the variation in morphology across three TPs before and after the mechanical activation procedure. For each group, the roundness and L/W ratio of 20 grains were calculated, of which the averages were used to quantify the particles. It is clear that after the first 30 min of grinding, from RAWTP to 0.5HTP, both the roundness and L/W increase, indicating that the morphology of particles changes from a sub-circular to a clavate shape.

As shown in Figure 4a, during the grinding process, when touched by and extruded between multiple grinding media, or between grinding media and tank body, an extrusion force is applied to the TP grain. The external, linearly applied force causes the particles to break in half or into pieces along the structural plane of the crystal lattice, and the particle size of TP also decreases significantly (Figure 3). Considering the nondeterminacy and unpredictable growth of crystal lattice, the TP grains tend to be much more irregular, as indicated by the increasing values of roundness and L/W ratio.

However, after the second 30 min, the values of roundness and L/W ratio decrease sharply, and the shapes of particles tend to return back to sub-circular again, as seen in Figure 4b. This phenomenon is highly consistent with the specific surface area and particle size discussed in Section 3.1.1. As a result of dimensional disparity between activated TPs and grinding media, the extrusion force between grains is much more attenuated, and the friction forces perpendicular to the contact surface of the particles dominate in this process, both between TP and grinding media and among the TP grains. Hence, as is confirmed by the distributive curve, the particle size of TP grains does not decrease dramatically, but the proportion of small-scale particles increases substantially. This phenomenon represents that the rough and angular edges of particles were stripped, as characterized by the declines in roundness and L/W values.

#### 3.1.4. Phase Assemblages

As well as improving the variation in dimension and morphology, mechanical activation has been reported to chemically improve the phase assemblages of raw materials. Figure 6 shows the mineral composition variation of TP at the different grinding times. The analysis result shows that with the increasing grinding time, the diffraction peak intensity of quartz declines, which indicates that the crystallinity of quartz in TP declines, and SiO_2_ gradually was translated from quartz to an amorphous state. Similar to morphology, the crystallinity of SiO_2_ dramatically decreases during the first 30 min. According to previously reported research, the pozzolanic reactivity of tailing and quartz powder is mainly attributed to the content of amorphous materials [49]. This is consistent with the results of a study by Cheng et al. [36], who reported that increasing grinding time increased the compressive strength of siliceous tailing-based concrete. However, the effect of changed particle size and morphology is not discussed deeply, and the pozzolanic reactivity of siliceous powders is supposed to be improved. Han et al. [29] conducted an experiment and reported that mechanically activated siliceous tailing powder still demonstrates low pozzolanic behavior, and the physical effects of dimensional size and particle shape play a more dominant role in the hydration of the composite binder. Hence, in order to elaborate on the detailed mechanism of mechanical activation, a series of pozzolanic reactivity tests must carried out.

#### 3.1.5. Calorimetry Test of Hydration Reactivity

In order to specifically evaluate the pozzolanic reactivity behavior of grinded TPs, a group of heat evolution rate tests were conducted, the results of which are reported in Figure 7.

Considering that heat related to the dissolutions of calcium hydroxide and calcium carbonate is negligible, the change in calorimetry could be mainly attributed to the dissolution of TPs, along with the reaction heat of TP particles and calcium hydroxide [50]. Moreover, the addition of sodium hydroxide could also improve the alkali phenomenon and accelerate the pozzolanic reaction of testing materials.

It can be clearly seen that the calorimetric curves of the three TPs are almost overlapping; the gap between them is smaller than 5%. This indicates that although activated, 0.5HTP and 1.0HTP do not represent a significantly higher pozzolanic behavior than that of inactivated RAWTP, meaning the effect of mechanical activation on the mineralogy characteristics of TP is exceedingly limited, even under an alkali condition. This result shows that the majority of the quartz in TP is not broken and depolymerized absolutely, in line with the mineral analysis of TP. Hence, the incorporation of TPs as an SCM does not influence pozzolanic behavior much, but may play a more dominant role in filling and improving the microstructure.

### 3.2. Air Content

Air content is an important parameter value in fresh slurry, which influences the workability and micro-pore growth of cementitious composites. The air contents of various blended paste specimens are illustrated in Figure 8.

It could be concluded that, compared with PC, all specimens containing TP demonstrate a higher air content, which indicates that the incorporation of TP probably deteriorates the pore structure of blended paste. This result could be ascribed to the different air-absorbed behaviors of cement and TP particles. Following the water film thickness theory (WFT) [51], the water film on the surface of TP particles would have much more air capacity during the mixing procedure, subsequently affecting the pore structure of the specimen.

It is worth noting that the air content of the fresh mixture is also enhanced with the increased grinding time. The air content is in good accordance with the workability, and a fresh mixture containing much more air characterizes poorer workability, which is adverse to the application of cementitious composites [52]. The larger specific surface area increased by grinding time results in a higher water absorption rate, a thicker WFT, and finally, a higher air content. However, as shown in Table 2, although the increasing percentage of specific surface area between 0.5HTP and 1.0HTP is large (51.47%), the gap between the two air contents is relatively low (2.54%). This result is significantly different from the relationship between RAWTP and 0.5HTP, which may be due to the difference in morphology between the two TPs.

Rod-shaped particles (such as those in 0.5HTP) accumulated in fresh slurry tend to form more triangular stereoscopic mesoscopic pores, which capture more free water and air bubbles, increasing the air content. Nevertheless, nearly spherical particles (RawTP and 1.0HTP) could exhibit a better packing model, ensuring that the distribution of free water in the slurry is more uniform, avoiding the concentration of pore water, and alleviating the increase in air content.

### 3.3. Setting Time

Figure 9 illustrates the setting properties of pastes with and without TPs. With the incorporation of TP replacing the partial cement content, both the initial and final setting time are extended critically. This result is highly consistent with the literature by Yao et al. [35], which has also reported a similar testing result regarding the effect of mechanically activated iron ore tailing on the setting time of blended cement. They reported that the initial and final setting times were extended by 51.7% and 44.4%, respectively, as the dosage of TP reached 30 wt%. This is mainly due to the dilution effect of cement. Zhao et al. [32] illustrated that some low pozzolanic SCMs can represent a nucleation effect and provide space for the hydration of cement grains. However, since the dramatic dilution of cement content still dominates, the nucleation effect of SCM is unable to offset this negative effect. It should also be noticed that, with the increasing activation period, both the initial and final setting time decrease. Especially for final set property, the setting time declines more remarkably, while the initial set represent relatively poor. This phenomenon could be explained as the nucleation effect divergence due to the enhancing specific surface area of TPs. Chen and Kwan [53] reported that an increase in the surface area of SCM would provide more nucleation points for cement hydration, and the hydration kinetics would be accelerated. Furthermore, the water absorption rate of SCM is also a dominant factor influencing the hydration kinetics, which are also related to the specific surface area of TP particles [51,54,55]. Meanwhile, the setting durations (the length of time from initial setting to final setting) sustained for 80, 83, and 74 min for RawTP, 0.5HTP, and 1.0HTP, respectively. It is clear that 0.5HTP enjoys the longest duration for setting, indicating that the morphological characteristics also have a bearing on the setting behavior of blended cementitious binder. Much like for air content, cumulation-forming pores would trap some of the free water, which could be seen as “trapped water” (Figure 10). As a result, the relative w/b ratio would decrease and deteriorate the hydration process eventually.

Overall, Zhu et al. [34] illustrated that a sufficient setting time and a shorter final setting time, i.e., a shorter setting duration in this study, is highly favorable in the construction field. Correspondingly, the nearly spherical and ellipsoidal shape of SCMs is beneficial to the setting behavior of cementitious binder, and much more fit for the construction field.

### 3.4. Hydration Reaction Kinetics

To investigate the impact mechanism of SCMs’ morphological properties on the hydration behavior of blended cement, the overall process of cementitious binder hydration was tested via isothermal calorimetry. Following reports by Marchon et al. [56] and Zhao et al. [13], the whole hydration procedure can be divided into five steps (Figure 11a): (1) the wetting of the cement surface and the fast dissolution of anhydrous phases; (2) sudden slowdown of the reaction and the induction period; (3) the massive precipitation of C-S-H and CH and the second acceleration period; (4) the second deceleration; and (5) the period of low activity due to the slow diffusion of species. Across these five steps, three main exothermic peaks are observed in all blended pastes. Peak Ⅰ is generally understood as the heat release caused by the wetting and dissolution of cement particles at this stage. Peak Ⅱ and Ⅲ correspond to the hydration heat flow of C_3_S and C_3_A, respectively.

From Figure 11a,b, the PC paste represents the highest hydration rate and total hydration heat. With the incorporation of TPs, both the hydration rate and total heat flow dramatically decrease. Much like with setting time, the decline of hydration behavior is mainly attributed to content loss and the dissolution effect of cement. This result means that TP has very low or no hydration reactivity at an early age, leading to the lower amount of hydration heat from blended cement. The heat flow rate of PC paste was highest during Step Ⅲ and Ⅳ and declined with the addition of TP. As discussed above, in these two steps, the hydration is mainly attributed to the mass of cement. The dilution effect of TP reduced the hydration heat of binders, leading to the retard during the second acceleration period, which is in line with the results from Goulart et al. [57].

Comparing the three blended binders of TPs with different activation times, the slurry with 1.0HTP represented the highest hydration rate and total hydration heat. This result is consistent with the results of a previous study by Han et al. [29]. Zhao et al. [58] also reported the hydration behavior of blended cement with wet-ground SCMs. Compared with RAWTP and 0.5HTP, the larger particle specific surface area of 1.0HTP provided many more nucleation sites. Meanwhile, the high water absorption rate of 1.0HTP is also a potential factor improving the hydration procedure.

Some researchers also assume that the wet milling process could dramatically accelerate the pozzolanic reactivity of silicon and aluminum SCM, which can also be seen in the calorimetry curve [59,60,61]. However, the response to heat flow results are shown in Figure 7, and the improvement in pozzolanic reactivity for all TPs is almost negligible. Therefore, the modification of hydration peak could be explained by the effect of the physical properties of TP caused by the grinding process, but not by the influence of chemical change. In comparison among the hydration heat flow rate curves, the hydration properties of the blended cementitious binders are strongly connected to the morphological and dimensional characteristics of the three TPs. In detail, the curves of 0.5HTP and 1.0HTP represent a similar changing tendency, where Peak Ⅲ dominates and Peak Ⅱ is relatively inferior. For the value order of Peak Ⅱ, i.e., heat flow from the reaction of C_3_S, 0.5HTP demonstrates lowest value, far lower than that of RAWTP and 1.0HTP. This result means that the clavate shape of TP deteriorates the hydration of C_3_S. On the contrary, the value order of Peak Ⅲ is proportional to specific surface area. This result seems to be due to the distinct particle morphology of the three TPs. As shown in Figure 10a,b, the particle shape differences between the TPs would result in different packing models. For 1.0HTP, the round shape particles could result in a dense packing model, and the water between the particles is free to react with cement (Figure 11a). However, the clavate shape of 0.5HTP particles may build a triangle skeleton (Figure 11b). Due to the WFT, water was trapped in the triangle, causing difficult reacting with cement particles, which may potentially account for the limiting of the C_3_S mineral phase.

### 3.5. Compressive Strengths

The compressive strength evolution of the pastes is given in Figure 12. Different modifications to the compressive strength behavior caused by the addition of various grinded TPs can be observed. No matter how long the curing time was, PC always showed the highest compressive strength, while all the specimens incorporating TPs showed a decline. The deterioration of compressive strength could be attributed to the dilution of the cement, which is much more conspicuous with an early curing. The dilution of cement resulted in fewer hydration products and poorer bonding effectiveness with the aggregate, which led to a lower strength of paste and poorer interface transition zone [62]. Despite the nucleation effect of TPs accelerating the hydration of cement, the dilution effect due to the decreased content of cement is much more serious. The nucleation sites supplemented by TP cannot cover the loss of C-S-H from C_3_S and C_2_S, eventually leading to the decrease in compressive strength. At the same time, Cheng et al. [36] used the dry milling method to stimulate the pozzolanic reactivity of TP and prepare concrete. Although activated, the compressive strength of concrete where cement w replaced by TP still decreases, especially at an exceedingly higher replacement degree. Moreover, this decline in compressive strength would modestly decrease after curing for 60 days. As previously reported, most SCMs often show low pozzolanic reactivity and react poorly to cementitious composites at an early age, but they mainly fulfil the role of filling [63,64]. Hence, mechanically activated TPs still showed a lower reactivity in the first 28 days.

On the other hand, the compressive strengths of pastes increased with the grinding time: 1.0HTP represented the highest value, and RAWTP enjoyed the lowest. It seems that the pozzolanic reactivity of TP is improved through the mechanical activation method. However, the pozzolanic testing result in Figure 7 suggests that the grinding procedure shows limited improvement. As a result, the mechanism of improving paste strength will be further discussed in hydration product and micropore analysis.

### 3.6. Hydration Products

To investigate the early pozzolanic reactivity of mechanically activated TP, Figure 13 shows the X-ray diffraction pattern of paste specimens after curing for 28 days. The main minerals of paste include the ettringite and portlandite from cement hydration, quartz from tailing, and C_3_S from unhydrated cement. It can be concluded that, except for the incorporated quartz, the addition of TP shows little effect on the hydration type, but only influences content. With the addition of TP, the content of C_3_S (29.5° and 32° 2θ) dramatically decreased. This phenomenon could be explained by two causes: the dilution effect of cement caused by the decreased content, and the nucleation effect of TP [65,66], which align with the hydration heat result. However, the effect of dilution performs much more than that of nucleation, finally leading to the decreases in C-S-H (29.5° 2θ) and CH (17° and 34° 2θ).

Moreover, the intensity of C_3_S in the RAWTP sample as higher than that of 0.5HTP and 1HTP, which indicates that the lower specific surface area represents a worse nucleation effect. However, this effect is very inconspicuous compared with the change in specific surface area. Furthermore, the continuous grinding of TPs hardly affect the type or content of hydration products. It can be observed from Figure 13 that grinding time shows almost no impact on the intensity of CH peak, which is generally supposed to relate to the pozzolanic reactivity of SCM, as shown in Equation (2) [38]. Considering that XRD test is a semi-quantitative analysis, further discussion is needed.
Ca(OH)_2_ + SiO_4_^4−^ → C-S-H(2)

### 3.7. Thermogravimetric Analysis

Figure 14 shows the DTG results of cementitious pastes without and with TP after curing for 28 days. There are mainly three descending peaks, at around 30~220 °C, 420~500 °C, and 640~740 °C, mainly corresponding to the dehydration of hydrates (AFm and C-S-H) [55], and the dihydroxylation reaction of CH [50] and CaCO_3_ [67], respectively. All of the pastes with TP showed lower hydrates contents than that of PC, due to the fact that cement is greatly diluted due to the replacement with TPs. The thermogravimetric results of the three specimens and DTG curves are almost identical. These also indicate that mechanical activation barely ameliorates the effect of TPs on the hydrates of cementitious composites, and prove that the major compressive strength enhancing and influence mechanism of TP is the micro filling effect. After curing for 28 d, 0.5HTP represents the lowest value of the peak corresponding to AFm and CSH, which signifies the highest content of hydrates and disagrees with the compressive strength result. This could be explained by the “water impounding effect”, as shown in Figure 10, where part of the water is not chemically reacted with inside hydrates, but physically bounded on the surface of TP particles. Hence, 0.5HTP shows the evident peak.

### 3.8. Pore Structure

Figure 15 illustrated the effect of different TPs on the pore structure of cementitious pastes. According to previous studies, the number of hydration products is the main influencing factor on microstructure [68,69,70]. Due to the temporary curing time (28 days), the still portion of the cementitious materials does not hydrate (Ca_3_SiO_5_ in Figure 13); therefore, the amount of C-S-H was also low, leading to all of the specimens representing poor microstructure, of which the pore diameters were almost higher than 100 nm [71]. From Figure 15a, without and with various TPs, the pore diameter concentrated on a different size. As discussed in Section 3.4, the dilution effect of cement and poor nucleation of RAWTP and 0.5HTP leads to the lower number of products, representing the larger pore diameter when compared with PC and 1.0HTP. At the same time, the divergence in particle size is also an important factor. As seen in Figure 2, the particle size distribution order is RAWTP > cement > 0.5HTP > 1.0HTP, which is consistent with the MIP test result, indicating that the coarser grains cannot fill and lead to bigger pore diameters.

In Figure 1 and Figure 15b, with the increasing grinding time, the cumulative pore volume of paste decreases, while that of PC remains between 1.0HTP and 0.5HTP. This aligns with the results of the compressive strength test (Figure 12). The incorporation of a finer particle (1.0HTP) plays a great role in the filling effect, making the microstructure denser. In contrast, coarser particles (e.g., RAWTP) lead to a worse packing and lower compressive strength. It is worth noting that, despite representing similar particle size distributions and packing models, the cumulative pore volume of 0.5HTP is higher than that of PC. Except for the effect of the number of hydration products, as shown in Figure 10b, the clavate shape possibly results in more holes caused by the triangle packing, finally leading to a looser packing model. The trapped water among the TP grains would be vaporized during the curing, forming a series of micro or macro pores.

### 3.9. Autogenous Shrinkage

Figure 16 shows the measured autogenous shrinkage of specimens after being cured for 3 days. The autogenous shrinkage value of the PC paste represents the highest value, which is very harmful to the development of mechanical behavior and durability. A higher internal tensile stress of microstructure would be created due to the higher autogenous shrinkage of the paste [72]. Considering the low tensile resistance of cementitious material, this internal tensile stress would induce the generation of cracks, which would finally deteriorate the pore structure and mechanical strength [73]. Hence, it is necessary to keep the autogenous shrinkage of cementitious paste as low as possible. Compared with PC, the total autogenous shrinkages of specimens decreased by 27.4%, 18.9%, and 17.9% after 3 days with the incorporation of RAWTP, 0.5HTP, and 1.0HTP, respectively. These results mean that the addition of TP as a kind of SCM could dramatically decrease the total autogenous shrinkage of cementitious paste [73]. This is consistent with other reports that the addition of SCMs could relieve the early shrinkage of cementitious composites [74]. A previous study has reported that the self-desiccation phenomenon is considered to be the main driving force behind autogenous shrinkage. In Figure 11b, all three different TPs decreased the total hydration heat and inhibited the early hydration process of cement, which weakened the self-desiccation phenomenon and led to a lower autogenous shrinkage [48].

Meanwhile, the coarser the particle size of TP grain, the lower the autogenous shrinkage. This result can be explained by the interaction of two aspects. Wang et al. [48] reported that a lower water-to-binder ratio results in a higher autogenous shrinkage. In this study, the w/b ratio was fixed at 0.5. However, according to the WFT, a higher specific surface area of powder grains means a higher water absorption rate. Therefore, the finer particle, i.e., 1.0HTP, will lock more free water in the mixture and decrease the relative water-to-cement ratio (w/c ratio). At the same time, the volumetric percentage of pore between 5 and 50 nm is also a factor affecting the autogenous shrinkage. Following the result of the MIP test in Figure 14, with the increasing grinding time, pores whose diameters were between 5 and 50 nm were enhanced. Li et al. [74] reported that the larger this pore volumetric percentage is, the more significant the capillary effect is, and the higher the autogenous shrinkage is. Due to both of these two factors, the autogenous shrinkage of paste inclined with the increasing grinding time of TP. Nevertheless, the autogenous shrinkage value of 1.0HTP is still lower than that of PC. Hence, as a replacement for cement, the incorporation of mechanically activated TP can reduce the autogenous shrinkage effectively, which is beneficial to the development of microstructure.

## 4. Conclusions

This study investigated the effect of the wet grinding method on the mechanical and chemical properties, especially the particle morphology, of siliceous iron ore tailing powder. In the meantime, cementitious pastes with different TPs were developed and compared with those which used only cement. Based on the testing results and discussions, the following conclusions can be drawn:(1)Throughout the grinding period, the particle size of TP gradually declines, and the decline range also decreases. The shapes of powder particles turn from sub-circular to clavate and then back to sub-circular, indicating that the grinding time significantly affects the morphology of particles.(2)The incorporation of TP delays the setting and hydration behavior of blended cement, and reduces the hydration product content. However, the further mechanically activated particles can play a role in nucleation and accelerate the hydration of cement particles. The mechanical activation of TP could significantly increase the compressive strength of blended cement pastes using TP, but these values are still lower than that of cement.(3)Autogenous shrinkage decreases with the addition of TP but increases with more grinding time. This indicates that the addition of TP has a positive effect on the early autogenous shrinkage of paste and relieves the growth of micro-crack.(4)This study discussed the morphological effect of TP on the performance of blended cemented paste. The result indicated that this morphological property significantly affects the hydration procedure, strength, and pore structure, but hardly influences the hydration products. However, the impact mechanism of mechanical grinding on other SCMs is still limited, and a further study is needed.

## Figures and Tables

**Figure 1 materials-16-01638-f001:**
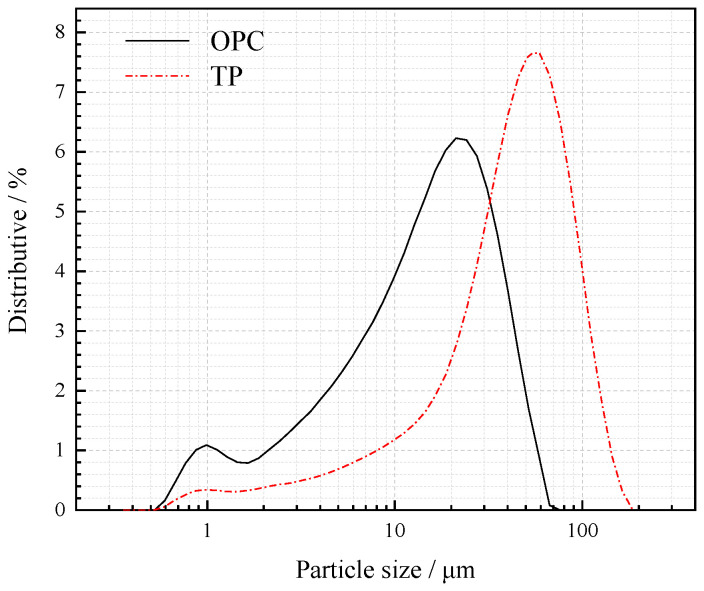
Particle size distribution comparison among TPs and cement.

**Figure 2 materials-16-01638-f002:**
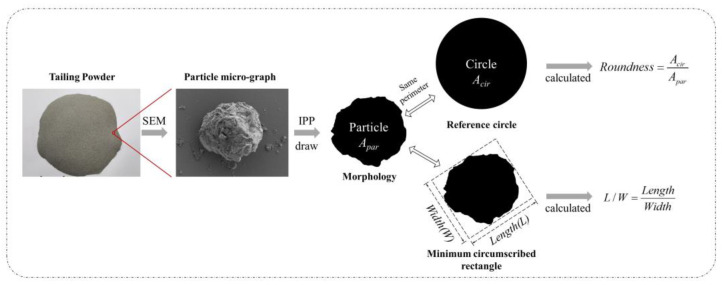
Schematic diagram of calculating the roundness of TP.

**Figure 3 materials-16-01638-f003:**
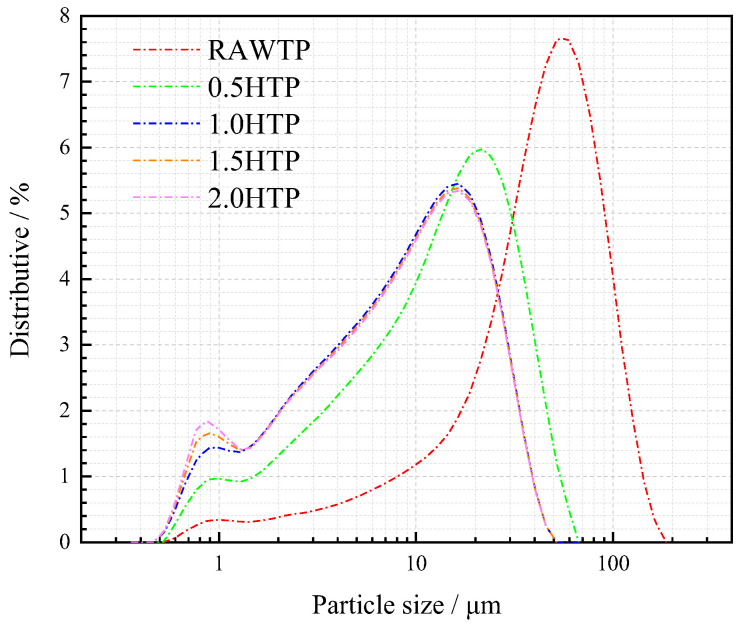
Particle size distribution variation of TP during mechanical activation.

**Figure 4 materials-16-01638-f004:**
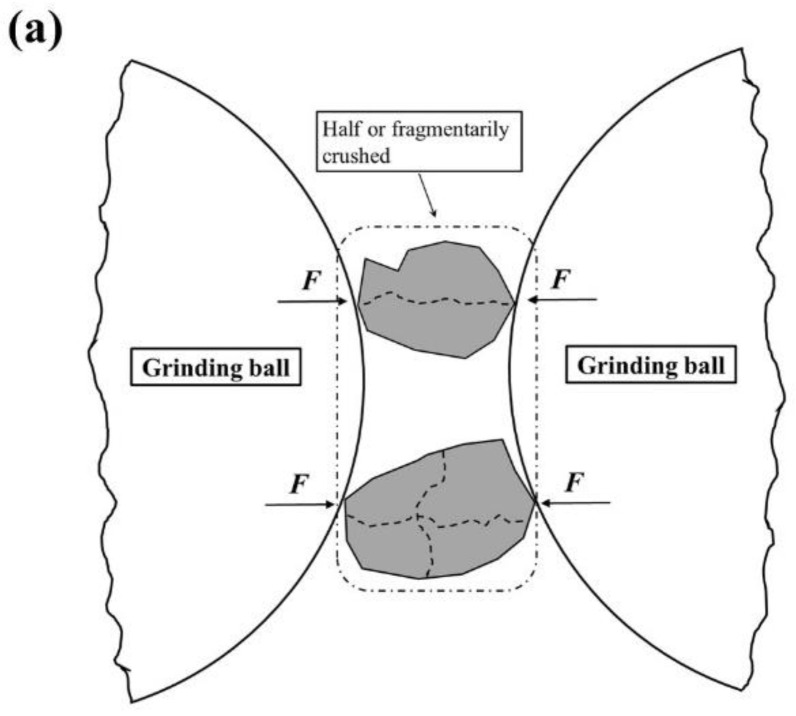
Schematic diagram of tailing particle damage during (**a**) Stage 1: Raw–0.5 h (**b**) Stage 2: 0.5–1 h.

**Figure 5 materials-16-01638-f005:**
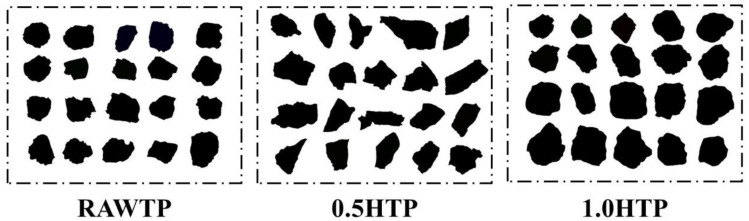
The morphological patterns of RAWTP, 0.5HTP, and 1.0HTP.

**Figure 6 materials-16-01638-f006:**
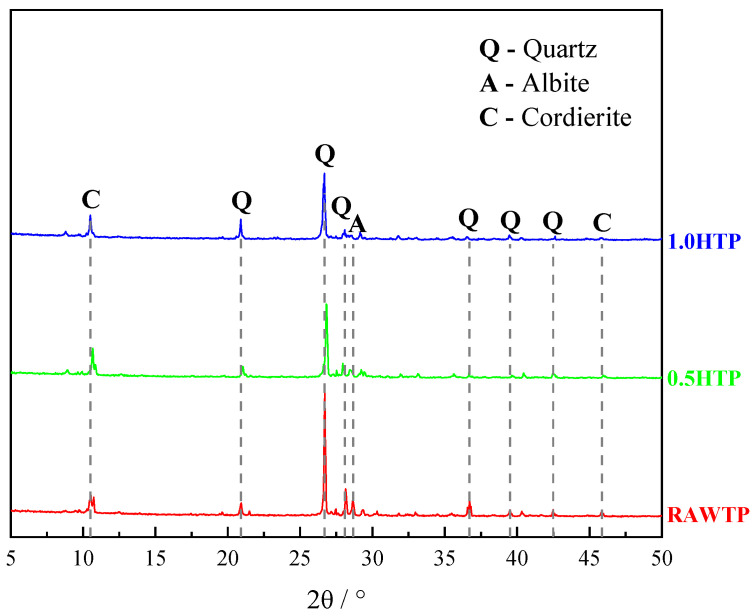
Mineral composition variation of TP during mechanical activation.

**Figure 7 materials-16-01638-f007:**
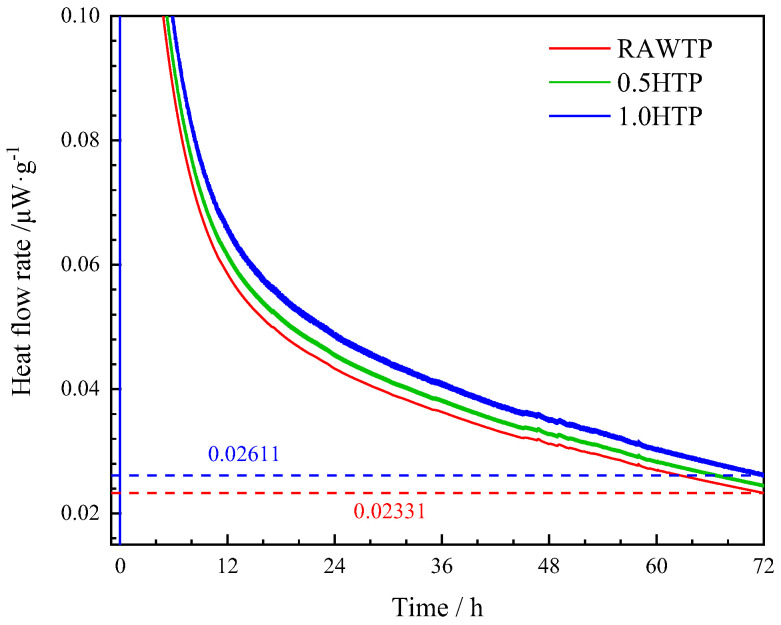
Calorimetry test results of the TPs’ pozzolanic reactivity.

**Figure 8 materials-16-01638-f008:**
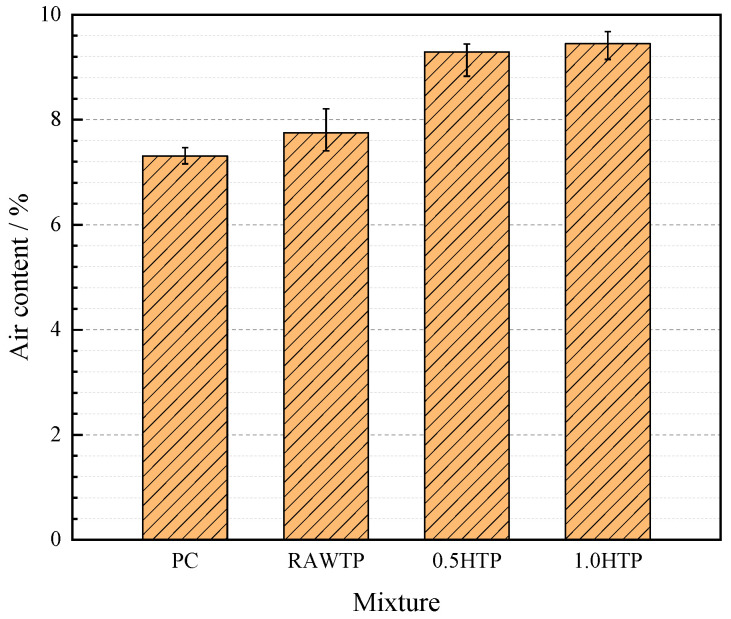
Air content of fresh cementitious pastes.

**Figure 9 materials-16-01638-f009:**
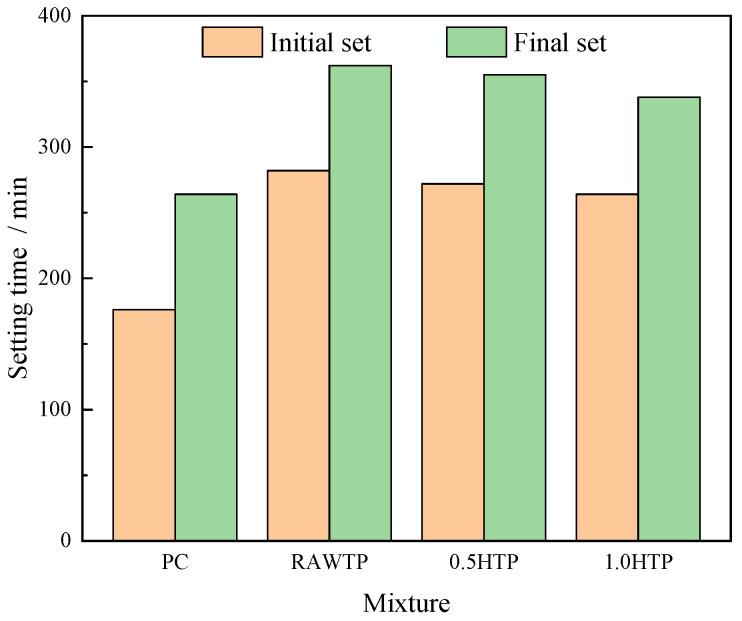
Setting times of cementitious pastes.

**Figure 10 materials-16-01638-f010:**
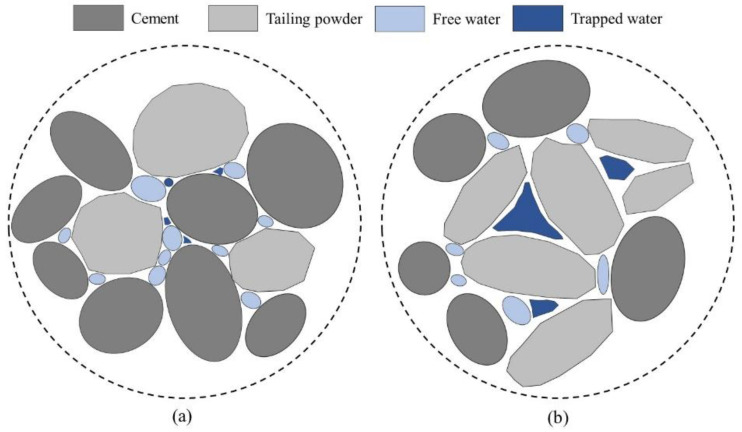
Schematic diagram of particle packing structure. (**a**) RAWTP or 1.0HTP; (**b**) 0.5HTP.

**Figure 11 materials-16-01638-f011:**
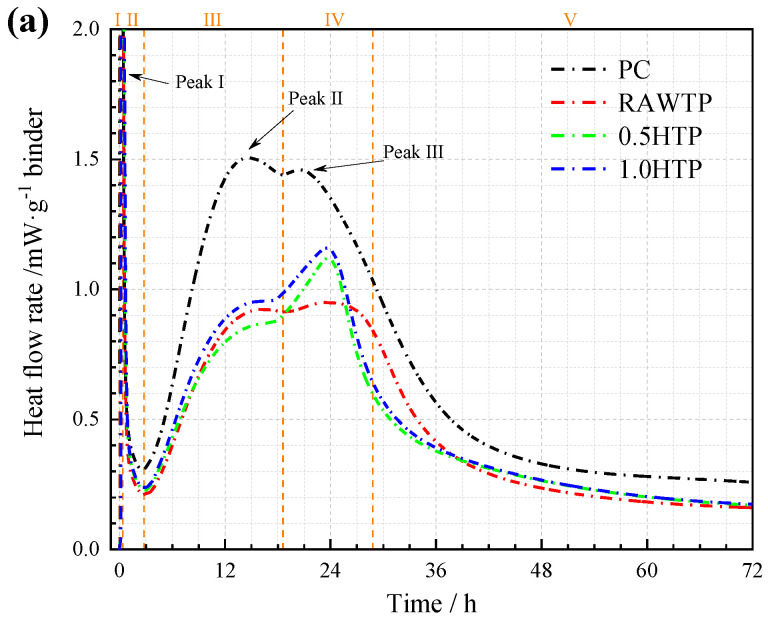
Calorimetry testing result of cementitious binder with and without TPs: (**a**) hydration heat rate; (**b**) cumulative hydration heat.

**Figure 12 materials-16-01638-f012:**
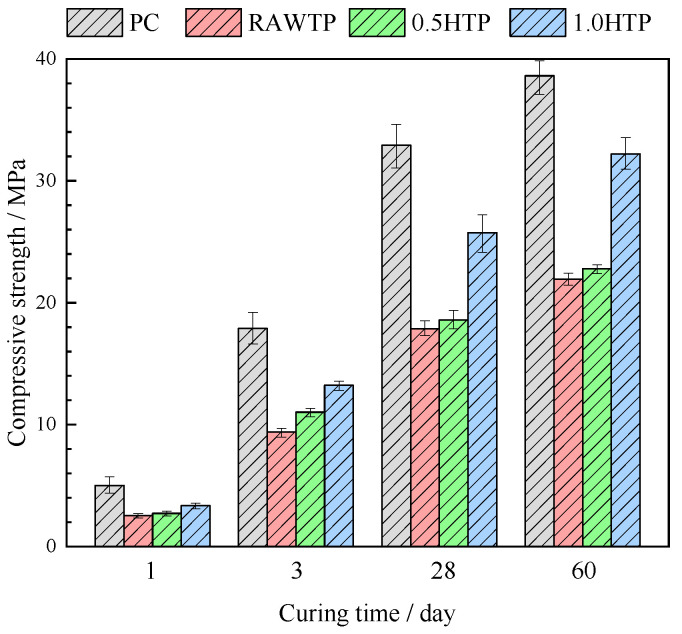
Compressive strengths of cementitious pastes.

**Figure 13 materials-16-01638-f013:**
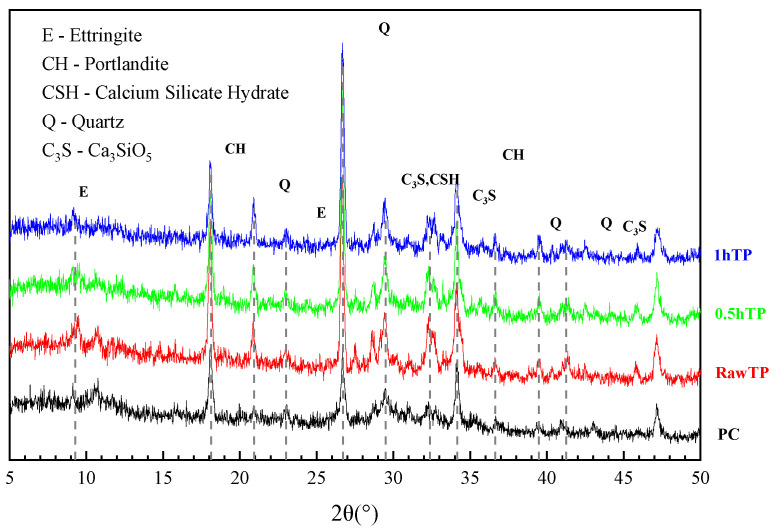
XRD results of cementitious pastes.

**Figure 14 materials-16-01638-f014:**
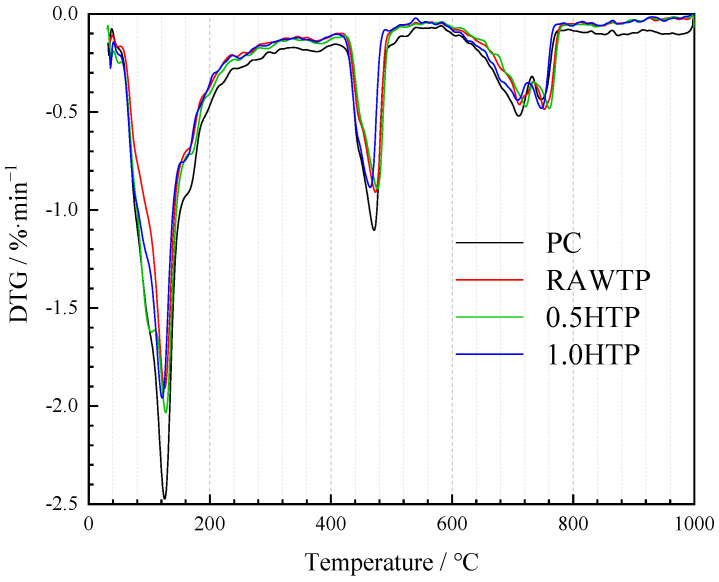
DTG result of cementitious pastes.

**Figure 15 materials-16-01638-f015:**
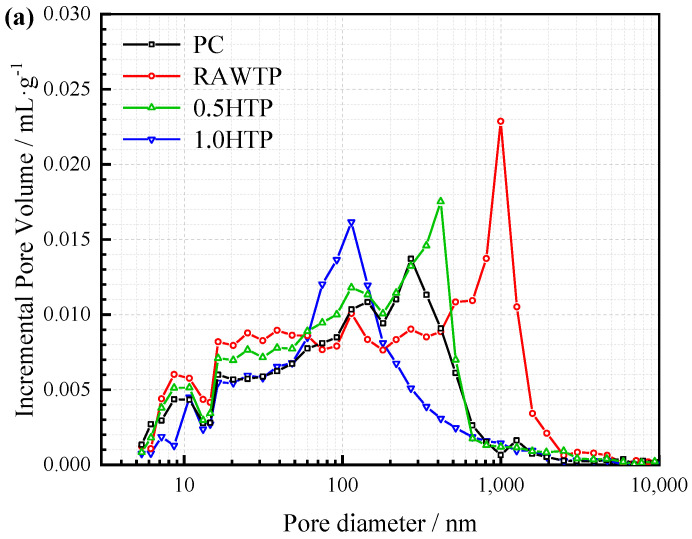
Incremental (**a**) and cumulative (**b**) pore volumes of cementitious pastes.

**Figure 16 materials-16-01638-f016:**
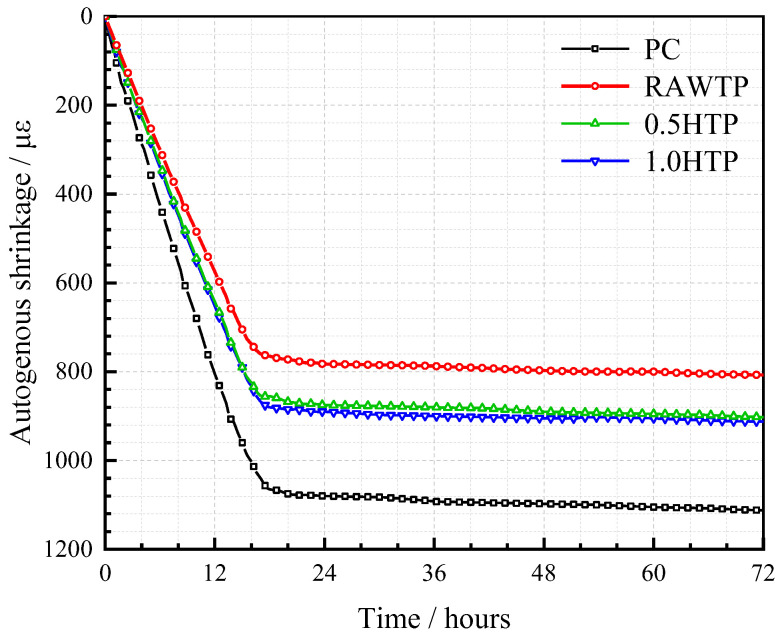
Autogenous shrinkage results of cementitious pastes.

**Table 1 materials-16-01638-t001:** The chemical composition of the cement.

	SiO_2_	CaO	Al_2_O_3_	MgO	Na_2_O	K_2_O	SO_3_	Fe_2_O_3_	LOI	BET Surface Area
OPC	20.19	63.03	5.11	1.72	0.10	0.32	1.19	2.11	2.14	1.9225 m^2^·g^−1^
TP	69.08	5.05	4.74	6.06	0.39	0.34	0.48	8.88	0.89	0.4437 m^2^·g^−1^

**Table 2 materials-16-01638-t002:** The physical properties of TP after mechanical activation.

	d_10_/μm	d_50_/μm	d_90_/μm	BET Surface Area
RAWTP	15.9	47.2	98.6	0.4437
0.5HTP	2.40	14.4	36.4	1.6595
1.0HTP	1.67	9.57	26.4	2.5136
1.5HTP	1.45	8.68	24.1	2.5611
2.0HTP	1.28	8.68	24.1	2.5784

**Table 3 materials-16-01638-t003:** The morphological parameters of TP during grinding.

Group	RAWTP		0.5HTP		1.0HTP	
	Roundness	L/W	Roundness	L/W	Roundness	L/W
1	1.242	1.06	1.4717	1.3	1.3074	1.2
2	1.276	1.3	1.4984	1.89	1.2282	1.14
3	1.389	1.66	1.4596	1.51	1.3055	1.22
4	1.327	1.26	1.3298	1.91	1.2445	1.16
5	1.326	1.34	1.3495	1.78	1.1731	1.23
6	1.364	1.29	1.6061	1.56	1.2799	1.18
7	1.326	1.28	1.4257	1.75	1.2023	1.19
8	1.373	1.31	1.4604	1.64	1.2394	1.14
9	1.353	1.43	1.6281	1.42	1.2451	1.19
10	1.309	1.42	1.6378	1.63	1.2815	1.26
11	1.286	1.35	1.6657	1.55	1.3295	1.16
12	1.315	1.29	1.4409	1.94	1.3045	1.16
13	1.307	1.42	1.4628	1.46	1.2811	1.21
14	1.317	1.32	1.4515	1.65	1.3347	1.16
15	1.275	1.26	1.4589	1.66	1.3073	1.25
16	1.262	1.31	1.5299	1.49	1.2389	1.16
17	1.331	1.25	1.4598	1.73	1.3823	1.17
18	1.342	1.25	1.3429	1.86	1.2645	1.24
19	1.397	1.43	1.3646	1.53	1.2625	1.26
20	1.318	1.24	1.474	1.88	1.3338	1.23
Average	1.322	1.32	1.4759	1.65	1.2773	1.2

## Data Availability

Not applicable.

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
