# Peer review of "Effect of Particle Size and Morphology of Siliceous Supplementary Cementitious Material on the Hydration and Autogenous Shrinkage of Blended Cement"

_materials, 2023, doi:10.3390/ma16041638_

Round 1
Reviewer 1 Report
Dear authors, please refer to the attached for the comments. Thank you

Author Response
We are very grateful for your constructive comments. As you mentioned, there are still some parts can be improved in abstract, figures and manuscript and conclusion. Following your kind advice, we have improved these parts and all modified sentences have been marked in yellow. The detailed modification could be observed in word document.

Reviewer 2 Report
- abstract needs some revision. Significant findings are required to be highlighted.
- Unclear image of Figure 3.
- On what information did you refer to illustrate Figure 4?
- no error bar in Figure 7?
- Why Figure 8a is missing trapped water?
- Correlation between paragraph or section need to be more details.
Author Response
Reviewer #1
Response:
We are very grateful for your constructive comments. As you mentioned, there are still some parts can be improved in abstract, figures and manuscript and conclusion. Following your kind advice, we have improved these parts and all modified substances have been marked in yellow.
- Abstract needs some revision. Significant findings are required to be highlighted.
Answer: Thanks for your kind suggestion. We have modified the abstract and highlighted the significant finding as follow:
“……should not be ignored. In this study, a wet-grinding method was carried out to pretreat an iron ore tailing powder (TP), and the divergency of pozzolanic behavior and morphology were investigated. Then the treated TPs were used to replace 30% cement preparing blended cementitious paste, and the impact mechanism of morphology on performance was studied emphatically.”
- Unclear image of Figure 3.
Answer: Thanks for your important remand. We are very sorry that we have made this mistake. We have revised this figure to make it clearer, as follow:
Figure 3. Schematic diagram of tailing particle damage during (a) Stage 1: Raw-0.5h (b) Stage 2: 0.5h-1h
- On what information did you refer to illustrate Figure 4?
Answer: Thanks for your suggestion. Figure 4 is a schematic diagram paragraph about how to determine the roundness of tailing powder, and the detailed calculation methods are listed.
We are very sorry that we did not demonstrated this figure well, and a supplementary information is added as follow:
The determination of parameters and calculating process of calculation is shown in Figure 4
- No error bar in Figure 7?
Answer: Thank you for your constructive comments. In present study, only one test was performed to determine the air content, hence there is no error bar. Considering this question, a triplicate testing process is carried out and the error bar is supplied. The modified figure is shown as follow:
Figure 7. Air content of fresh cementitious pastes
The modified testing method introduction is also revised in Section 2.5.1 as follow:
……suggested by Chen et al[37]. A triplicate testing procedure was performed and the average determined the air content value.
- Why Figure 8a is missing trapped water?
Answer: Thank you for your constructive question. Firstly we are very sorry that the number for this photograph is wrong and it has been modified right (Figure. 9 now). Besides, the trapped water in Figure. 9(a) is supplied, as follow:
Figure 9. Schematic diagram of particle packing structure. (a) RAWTP or 1.0HTP; (b) 0.5HTP
- Correlation between paragraph or section need to be more details.
Answer: We appreciate your constructive comments. As you mention, the correlation between paragraph and section is poor, and we have modified the manuscript, as follow:
On the other hand, the compressive strengths of pastes increase with the grinding time, that 1.0HTP represents the highest, and RAWTP enjoys the lowest. It seems that the pozzolanic reactivity of TP is improved through mechanical activation method. However, the pozzolanic testing result in Figure 6 suggests that the grinding procedure shows limited improvement. As a result, the mechanism of improving paste strength will be further discussed in hydration product and micropore analysis.
Moreover, the intensity of C3S in RAWTP sample is higher than 0.5HTP and 1HTP, which indicates that the lower specific surface area represents a worse nucleation effect truly. However, this effect is very unconspicuous compared with the change of specific surface area. Furthermore, the continuous grinding TPs hardly affect both type and content of hydration products. It can be observed from Figure 12 that grinding time shows almost no impact on the intensity of CH peak, which is generally supposed to the pozzolanic reactivity of SCM, as shown in Eq.2[60]. Considering that XRD test is a semi-quantitative analysis, further discussion is needed.
Ca(OH)2+SiO44-→C-S-H (2)

Reviewer 3 Report
1. Clearly indicate the novelty of the research as there are many studies available where different kind of SCMs are being used as replacement of cement.
2. Proof read whole paper as there are many grammatical and Sentence formation mistakes, as below:
· In this study, a type of iron ore tailing powder (TP) as a siliceous supplementary cementitious material is promising……………
· …….inhibit the hydration compared with sub-circle one
3. Line 118: How mechanical force can affect solid particles chemically?
4. What was test conditions for XRD and TG analysis?
5. Change the caption of Figure 2.
6. Figure 3 is blurred. Make it clearer.
7. Figure 8. Setting times of cementitious pastes, is not clear. Make it in different style in which IST and FST will be clear visible.
8. Use subscript properly in whole manuscript.
9. Correct Figure 10 caption. …with and TPs…
10. Line 500: Fig. 13 shows the DSC-TG results?? Where is DSC result ?
11. What was the synergy observed within the cementitious matrix; it is not clear anywhere. Authors should give the chemical reaction or some activation energy relation to validate this.
12. Results and discussion section is very weak. Some results such as Compressive strength, hydration, XRD, MIP etc. are discussed in very brief. Please explain more and cite the recent literatures.
13. Conclusion: What are the limitations of this study?
14. The authors could extend the discussion section by comparing their results with previously published studies.
15. The authors can also recommend some future research work.
16. Some relevant latest references should be added to the literature review section.
17. Conclusions need to be revised by only presenting the key findings of the present work.
Author Response
Reviewer #2
Response:
Your kind effort and time in processing our manuscript is much appreciated. We have read the fair comments from you. Please see our detailed response as below. The modified parts have been marked in yellow.
- 1. Clearly indicate the novelty of the research as there are many studies available where different kind of SCMs are being used as replacement of cement.
Answer: Thanks for your kind suggestion. The main novelty of this work is studying the effect of particle shape on the pozzolanic of tailing powder and blended cement. We have modified our abstract to make it clearer.
“……should not be ignored. In this study, a wet-grinding method was carried out to pretreat an iron ore tailing powder (TP), and the divergency of pozzolanic behavior and morphology were investigated. Then the treated TPs were used to replace 30% cement preparing blended cementitious paste, and the impact mechanism of morphology on performance was studied emphatically.”
- Proof read whole paper as there are many grammatical and Sentence formation mistakes, as below:
-In this study, a type of iron ore tailing powder (TP) as a siliceous supplementary cementitious material is promising……………
-…….inhibit the hydration compared with sub-circle one
Answer: We are deeply grateful to the reviewer for taking the time to provide quite valuable criticisms. The grammatical mistakes in manuscript have been modified, as follow:
“……and total heat, and compared with sub-circle one, the clavated particle could inhibit the hydration procedure.”
- Line 118: How mechanical force can affect solid particles chemically?
Answer: We truly appreciate your constructive comments. Based on the study by Cheng et al, the grinding process is always considered to be an activation method that can both influence physical and chemical property of material. Through grinding process, except for the decreased size of large particles, some bonds of mineral phase may break under the action of mechanical force, which is reported as “mechanochemically activated” ().
- What was test conditions for XRD and TG analysis?
Answer: We highly appreciate your valuable comments. We are very sorry that we have missed the testing process for XRD and TG analysis, and these has been supplied into manuscript as follow:
“……and ground into powder. After this, the mineralogical and quantitative properties of hydration products were determined through XRD and TG analysis under the room temperature.”
- Change the caption of Figure 2.
Answer: Thank you for pointing out. The caption of Figure 2 has been revised, as follow:
Figure 2. Particle size distribution variation of tailing powder during mechanical activation
- Figure 3 is blurred. Make it clearer.
Answer: Thanks for your important remand. We are very sorry that we have made this mistake. We have revised this figure to make it clearer, as follow:
Figure 3. Schematic diagram of tailing particle damage during (a) Stage 1: Raw-0.5h (b) Stage 2: 0.5h-1h
- Figure 8. Setting times of cementitious pastes, is not clear. Make it in different style in which IST and FST will be clear visible.
Answer: Thanks for your guidance. We have improved the quality of Figure 8 to make it more intuitive.
Figure 8. Setting times of cementitious pastes
- Use subscript properly in whole manuscript.
Answer: Thank you for kind pointing out. The subscripts in manuscript have been modified and marked in yellow.
- Correct Figure 10 caption. …with and TPs…
Answer: Thank you for kind guidance. The caption of this figure has been corrected, as follow:
Figure 10. Calorimetry testing result of cementitious binder with and without TPs: (a) Hydration heat rate; (b) Cumulative hydration heat
- Line 500: Fig. 13 shows the DSC-TG results?? Where is DSC result ?
Answer: Thank you for comment. It should be DTG result, and we have corrected this mistake.
“Figure 13 shows the DTG results of cementitious pastes without and with TP after curing for 28 days.”
- What was the synergy observed within the cementitious matrix; it is not clear anywhere. Authors should give the chemical reaction or some activation energy relation to validate this.
Answer: We highly appreciate your valuable comments. The synergy between cement and SCM comes from the reaction of calcium hydroxide produced from C3S and C2S hydrate with reactive aluminum or silicate in SCM. This reaction is also called pozzolanic reaction. The chemical equation is as follows:
Ca(OH)2+[SiO4]4-→C-S-H
In this work, Ca(OH)2 content is used as one of the factor to measure the pozzolanic reactivity of TP, and this reaction is also added to the manuscript.
- Results and discussion section is very weak. Some results such as Compressive strength, hydration, XRD, MIP etc. are discussed in very brief. Please explain more and cite the recent literatures.
Answer: Thank you for your kind suggestion. As you suggest, a further discussion on the impact mechanism of mechanical activation on tailing powder has been supplied in different section, as follow:
3.5. Compressive strengths
On the other hand, the compressive strengths of pastes increase with the grinding time, that 1.0HTP represents the highest, and RAWTP enjoys the lowest. It seems that the pozzolanic reactivity of TP is improved through mechanical activation method. However, the pozzolanic testing result in Figure 6 suggests that the grinding procedure shows limited improvement. As a result, the mechanism of improving paste strength will be further discussed in hydration product and micropore analysis.
3.6. Hydration products
Moreover, the intensity of C3S in RAWTP sample is higher than 0.5HTP and 1HTP, which indicates that the lower specific surface area represents a worse nucleation effect truly. However, this effect is very unconspicuous compared with the change of specific surface area. Furthermore, the continuous grinding TPs hardly affect both type and content of hydration products. It can be observed from Figure 12 that grinding time shows almost no impact on the intensity of CH peak, which is generally supposed to the pozzolanic reactivity of SCM, as shown in Eq.2[61]. Considering that XRD test is a semi-quantitative analysis, further discussion is needed.
Ca(OH)2+SiO44-→C-S-H (2)
- Conclusion: What are the limitations of this study?
Answer: Thank you for your question. This manuscript studied the effect of mechanical grinding on the morphology of tailing powder, and applies it into cement paste. The change of hydration products was then discussed. However, the effect of mechanical grinding on other common SCM, such as steel slag and clay, still need a further work.
- The authors could extend the discussion section by comparing their results with previously published studies.
Answer: Thanks for your advice. We have added our previous published studies in our manuscript, as follow:
It can be observed from Figure 12 that grinding time shows almost no impact on the intensity of CH peak, which is generally supposed to the pozzolanic reactivity of SCM, as shown in Eq.1[61].
- The authors can also recommend some future research work.
Answer: Thanks for your kind suggestion. We have supplied our future research in Conclusion section, as follow:
(4) This study discussed the morphological effect of tailing powder on the performance of blended cemented paste. The result indicated that this morphological property significantly affects the hydration procedure, strength and pore structure, but hardly influences the hydration products. However, the impact mechanism of mechanical grinding on other SCMs is still limited, and a further study is needed.
- Some relevant latest references should be added to the literature review section.
Answer: We appreciate your constructive comments. As you mentioned, some latest references have been added in Introduction section, and these literatures are as follow:
- Zhao, J.; Li, Z.; Wang, D.; Yan, P.; Luo, L.; Zhang, H.; Zhang, H.; Gu, X., Hydration superposition effect and mechanism of steel slag powder and granulated blast furnace slag powder. Construction & building materials. 2023, 366, 130101. http://dx.doi.org/10.1016/j.conbuildmat.2022.130101
- Zhao, Q.; Pang, L.; Wang, D., Adverse Effects of Using Metallurgical Slags as Supplementary Cementitious Materials and Aggregate: A Review. Materials (Basel). 2022, 15 (11), 3803. http://dx.doi.org/10.3390/ma15113803
- Li, G.; Zhou, C.; Ahmad, W.; Usanova, K. I.; Karelina, M.; Mohamed, A. M.; Khallaf, R., Fly Ash Application as Supplementary Cementitious Material: A Review. Materials (Basel). 2022, 15 (7), 2664. http://dx.doi.org/10.3390/ma15072664
- Sun, X.; Liu, J.; Zhao, Y.; Zhao, J.; Li, Z.; Sun, Y.; Qiu, J.; Zheng, P., Mechanical activation of steel slag to prepare supplementary cementitious materials: A comparative research based on the particle size distribution, hydration, toxicity assessment and carbon dioxide emission. Journal of Building Engineering. 2022, 60, 105200. http://dx.doi.org/https://doi.org/10.1016/j.jobe.2022.105200
- Conclusions need to be revised by only presenting the key findings of the present work.
Answer: Thank you very much for your advice. We have refined the conclusion, and the modified conclusion is as follow:
(1) During the same grinding period, the particle size of tailing powder gradually declines, and the decline range also decreases. The shape of powder particles turns from ellipse to clavate and then back to ellipse, indicating that the grinding time significantly affects the morphology of particles.
(2) The incorporation of tailing powder would delay the setting and hydration behavior of blended cement, and reduce the hydration product content. However, the greater mechanical activated grain can play the role of nucleation and accelerate the hydration of cement particles. The mechanically activated behavior could significantly increase the compressive strength of blended cement pastes, but still lower than that of cement.

Reviewer 4 Report
The work is interesting. The authors have done well work. Herewith are some comments that could improve the article's quality.
1- In line 108 correct "A type of P.O. 42.5 cement…"
2- Explain the standard used for preparing the mix design.
3- In line 152 explain the method procedures used for air content then mention the dependent standard
4- In line 161 the method procedure for calculating the heat of hydration then explain it with photo.
5- Explain the standard used for MIP testing.
6- In line 179 wrote " The pretreatment method of the slice was the same as that for XRD and TG" but the sample use for MIP test must be in piece not crushed like XRD and TG. The authors must explain that statement.
7- In line 218 illustrate " Figure 2. This is a figure. Schemes follow the same formatting"
8- Figure 8. is not clear for illustrating the initial and final setting time.
9- Check the English language and grammar in the whole of the manuscript.
10- It must be used the same references written in the introduction section in the discussion sections.
11- It must be used the references directly related to this manuscript.
For writing the references in text must be written after the author name directly the number of the reference such as " Wang et al. reported …." Must be corrected as " Wang et al. [45] reported….."
Author Response
Reviewer #3
The work is interesting. The authors have done well work. Herewith are some comments that could improve the article's quality.
Response:
Your kind effort and time in processing our manuscript is much appreciated. We have read the fair comments from you and the anonymous reviewer with a big interest. Please see our detailed response as below. The modified parts have been marked in yellow.
- In line 108 correct "A type of P.O. 42.5 cement…"
Answer: Thanks for your kind suggestion. We have corrected this mistake, as follow:
A P.O. 42.5 cement (OPC), provided by……
- Explain the standard used for preparing the mix design.
Answer: Thank you for your constructive comments. The standard for mix design in this study is follow Chinese Standard 17671-2021, ‘Test method of cement mortar strength(ISO method)’, where the supplementary cementitious materials replace 30 wt% of cement.
……0.5 simultaneously, following the Chinese Standard GB/T 17671-2021.
- In line 152 explain the method procedures used for air content then mention the dependent standard
Answer: We appreciate your constructive comments. The main procedure for determining the air content has been added in Section 2.5.1, as follow:
……by Chen et al[37]. The air contents of fresh pastes were measured following ASTM C231-17a using a Type-B meter, and calculated by using Eq. (1).
As = A1-G (1)
where As is the air content of each fresh mixture (%), A1 is the air content shown by the Type-B meter (%), and G is correction factor (%). A triplicate testing procedure was performed and the average determined the air content value.
- In line 161 the method procedure for calculating the heat of hydration then explain it with photo.
Answer: Thank you for pointing out. The hydration heat is directly tested by testing machine follow Chinese Standard Test methods for heat of hydration of cement (GB/T 12959-2008). And we think that the calculation of hydration heat has a weak relationship with this study. If you think that this calculation could greatly improve the quality of this manuscript, we are glad to revise it in the next review.
- Explain the standard used for MIP testing.
Answer: We appreciate your constructive comments. We are sorry that there is no specific standard for MIP testing, and this testing method has been already used in many literatures.
- In line 179 wrote " The pretreatment method of the slice was the same as that for XRD and TG" but the sample use for MIP test must be in piece not crushed like XRD and TG. The authors must explain that statement.
Answer: We truly appreciate your constructive comments. And we have revised the section as your kind suggestion as follow:
……was used. A slice cut from 28 d curing specimen was tested to determine the micropore structure, which was impregnated with isopropanol and dried for 24 hours.
- In line 218 illustrate " Figure 2. This is a figure. Schemes follow the same formatting"
Answer: Thank you for pointing out. The caption of Figure 2 has been revised, as follow:
Figure 2. Particle size distribution variation of tailing powder during mechanical activation
- Figure 8. is not clear for illustrating the initial and final setting time.
Answer: Thanks for your guidance. We have improved the quality of Figure 8 to make it more intuitive.
Figure 8. Setting times of cementitious pastes
- Check the English language and grammar in the whole of the manuscript.
Answer: We truly thanks for your kind suggestions. The language and grammar in this manuscript have been improved and marked in yellow.
- It must be used the same references written in the introduction section in the discussion sections.
Answer: Thanks for your kind suggestion, but we are very sorry that we could not fully understand this suggestion. We have already revised some references in the introduction and discussion sections, and all these have been marked in yellow.
- It must be used the references directly related to this manuscript.
For writing the references in text must be written after the author name directly the number of the reference such as " Wang et al. reported …." Must be corrected as " Wang et al. [45] reported….."
Answer: We are very grateful for your constructive comments. We have removed the references that is not directly related to this manuscript, and the format for all references has been revised following your suggestion, which is marked in yellow.

Round 2
Reviewer 2 Report
Accepted for publication
Author Response
Thank you for your affirmation on this study and thank you again for your valuable comments on our manuscript very much.

Reviewer 3 Report
The paper can be accepted now.
Author Response
Thanks very much for your affirmation on this study and thank you again for your valuable comments on our manuscript very much.

Reviewer 4 Report
The authors fixed the required mistakes but English language, grammar, and style are minor spell check required.
Author Response
Thanks very much for your affirmation on this study and thank you again for your valuable comments on our manuscript very much. We have checked and cleared the language mistake again, and the modified substances have been marked in yellow.
